# Development of Aldehyde Functionalized Iridium(III) Complexes Photosensitizers with Strong Visible-Light Absorption for Photocatalytic Hydrogen Generation from Water

Xiao Yao [1], Qian Zhang [1], Po-Yu Ho [1,†,‡], Sze-Chun Yiu [1] 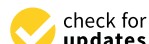, Songwut Suramitr [2,3], Supa Hannongbua [2,3,*] and Cheuk-Lam Ho [1,4,*]

1 Department of Applied Biology and Chemical Technology, The Hong Kong Polytechnic University, Hung Hom, Hong Kong, China
2 Department of Chemistry, Faculty of Science, Kasetsart University, Bangkok 10990, Thailand
3 Center for Advanced Studies in Nanotechnology for Chemical, Food and Agricultural Industries, KU Institute for Advanced Studies, Kasetsart University, Bangkok 10900, Thailand
4 PolyU Shenzhen Research Institute, Shenzhen 518057, China
* Correspondence: fscisph@ku.ac.th (S.H.); cheuk-lam.ho@polyu.edu.hk (C.-L.H.)
† Current address: Biological Inorganic Chemistry Laboratory, The Francis Crick Institute, 1 Midland Road, London NW1 1AT, UK.
‡ Current address: Department of Chemistry, King's College London, Britannia House, 7 Trinity Street, London SE1 1DB, UK.

**Abstract:** Four iridium(III) dyes functionalized with aldehyde functional group in the cyclometalating (CˆN) ligands, bearing either diethyl [2,2′-bipyridine]-4,4′-dicarboxylate or tetraethyl [2,2′-bipyridine]-4,4′-diylbis(phosphonate) anchoring groups, coded as **Ir1–Ir4**, are synthesized and explored as photosensitizers. The synthetic route is described and all of the complexes are characterized with respect to their electrochemical and photophysical properties. Density functional theory (DFT) calculation was used to gain insight into the factors responsible for the photocatalytic properties of **Ir1–Ir4** as effective photosensitizers for photocatalytic hydrogen generation. Relative to common iridium(III) dyes, such as **[Ir(ppy)₂(dcbpy)]⁺** (ppy = 2-phenylpyridine), the absorption spectra of our dyes are broader, which is attributed to the extended π-conjugation in their CˆN ligands. All of the new iridium(III) dyes were used as photosensitizers for visible-light driven hydrogen production by attaching to platinized TiO₂ nanoparticles (Pt–TiO₂) in the presence of sacrificial electron donor (SED) of ascorbic acid (AA) in a purely aqueous solution. A H₂ turnover number (TON) up to 5809 was demonstrated for 280 h irradiation. Complexes with tetraethyl [2,2′-bipyridine]-4,4′-diylbis(phosphonate) anchoring groups were found to outperform those with classical diethyl [2,2′-bipyridine]-4,4′-dicarboxylate, which may be one of the important steps in developing high-efficiency iridium(III) photosensitizers in water splitting hydrogen generation.

**Keywords:** aldehyde; photosensitizers; isoquinoline; hydrogen generation; iridium(III) complexes

## 1. Introduction

As there are growing concerns about the increasing global need for energy and the limited availability of fossil fuels, the investigation of the use of alternate energy sources has increased [1]. Being a renewable and carbon-free energy source, solar power has great potential for resolving the world's current energy problem [2]. Through the use of photocatalytic devices, it is possible to capture and convert solar power into many other kinds of usable energy [3–8]. Photocatalytic water-splitting using solar energy offers a sustainable approach to produce hydrogen fuel [9,10]. In 1972, Fujishima and Honda announced the development of the first light-driven water splitting hydrogen generation device based on TiO₂ [11]. As a result of this ground-breaking study, various water-splitting hydrogen generation systems were developed [12,13].

Iridium(III) cyclometalated complexes are attractive candidates as photosensitizers at TiO$_2$ to extend its absorption coverage, with outstanding photocatalytic hydrogen production performance. Bernhard et al. have investigated the iridium(III) metal complexes with the basic formula **[Ir(C^N)$_2$(N^N)]$^+$** (C^N = cyclometalating ligand, N^N = anchoring auxiliary ligand) for their uses in photochemical water-splitting hydrogen generation systems [2,14–16]. A TON of 800 was achieved with 50 μM of **[Ir(ppy)$_2$(bpy)]$^+$** as photosensitizers in a water:acetonitrile (1:1, *v/v*) solvent mixture [14,17–20]. Given that, curiosity has been piqued by iridium(III) complexes owing to their potential applicability as photosensitizers for hydrogen generation [21]. Iridium(III) complexes, with their 5*d* valence shell, display superior ligand-field stabilization energy, making them more stable during photocatalytic studies compared to other metalated dyes incorporating first- and second-row transition metals [2,22]. Instead of being a dissociative triplet metal centered ($^3$MC) state, the lowest energy triplet state for iridium(III) dyes is a triplet ligand centered ($^3$LC) state or a triplet metal-to-ligand charge transfer ($^3$MLCT) state, which usually encourages all of the achievable thermal deactivation pathways of the excited states, and therefore a longer lifetime of excited states in iridium(III) dyes can be achieved [2,22–26]. Moreover, physicochemical properties, such as tuning the energy bandgap by changing ligand molecular configurations, are viable strategies [27,28]. These favorable properties make iridium(III) complex an attractive candidate for hydrogen production via water-splitting. However, one disadvantage of iridium(III) complexes is their poor visible light absorbance [8,28–30], which can be improved by modifying their structures [31].

Structure-based modifications of cyclometalating and auxiliary ligands have been at the forefront of recent advances in iridium(III) photosensitizers, allowing for the tailoring of their electrochemical and photophysical properties [3,32]. Their light-harvesting capacity, electron injection efficiency, and stability can be enhanced by wisely selecting chromophores and meticulously constructing their molecular structure. Isoquinoline functional groups have been extensively applied in metalated dyes for water-splitting owing to their remarkable charge transfer ability [33–35]. While aldehyde moiety is usually employed to red-shift the absorption spectrum of metal complexes, which has also been exploited in iridium(III) compounds for targeting bio-related applications [36–40]. These approaches provide an attractive opportunity for developing highly efficient iridium(III) dyes for water-splitting systems.

Previous literature shows that anchoring groups will have a significant impact on the photocatalytic water splitting hydrogen generation performance [41,42]. Different anchoring groups, including carboxylate or phosphonate can be introduced in the bipyridine ligand of [Ir(C^N)$_2$(N^N)]$^+$-type dyes [42] to immobilize their linkages with the semiconductor [43]. However, it has been found that hydrolysis of the carboxylate anchoring sites reduces the efficacy of electron transport from photosensitizers to TiO$_2$ [44]. Compared to the carboxylate functional group, using the phosphonate functional group to anchor the dye to the TiO$_2$ surface improves the photocatalytic stability [44]. Hereunder, four iridium(III) complexes (**Ir1–Ir4**) bearing either diethyl [2,2′-bipyridine]-4,4′-dicarboxylate or tetraethyl [2,2′-bipyridine]-4,4′-diylbis(phosphonate) anchoring unit were synthesized as photosensitizers to facilitate light-driven water splitting hydrogen generation (Figure 1). Either thiophene-2-carbaldehyde-isoquinoline or thiophene-2-carbaldehyde-pyridine molecular was utilized as the primary structural unit to provide better molar absorptivity ($\varepsilon$) in the visible light region [38,45–50]. **[Ir(ppy)$_2$(dcbpy)]$^+$** was also synthesized as standard for comparing the light harvesting ability and water spitting hydrogen generation performance of our dyes. All of the iridium(III) dyes were measured electrochemically and photo physically. Their photocatalytic performances for hydrogen generation were then examined carefully through water-splitting. The structure-property relationship with various anchoring groups and the hydrogen production activity of the dyes are also discussed herein.

**Figure 1.** The chemical structures of iridium(III) dyes **Ir1–Ir4** and **Ir(ppy)₂(dcbpy)**.

## 2. Materials and Methods

### 2.1. Materials and Reagents

The regular Schlenk method was used for all of the reactions, which were conducted in a nitrogen environment. Prior to usage, glassware was dried in an oven overnight. Solvents were distilled using a suitable drying agent in a nitrogen environment. All of the chemicals were purchased from Sigma-Aldrich or Dieckmann and were employed without additional purification from their original form unless otherwise noted. With silica gel-coated aluminum plates purchased from Merck, the thin-layer chromatography was employed to monitor the progress of reactions. Dieckmann-purchased silica gel (230−400 mesh) was used in a column chromatograph to purify the products.

### 2.2. Instrumentation

$^1$H and $^{13}$C NMR spectroscopy were conducted using deuterated chloroform (CDCl$_3$) as the solvent on a Bruker Ultra-shield 400 MHz FT-NMR spectrometer (Figures S1–S10). Tetramethylsilane (TMS) was used as the chemical shift calibration reference. Mass spectrometry (MS) analyses were performed using an Agilent 6540 mass spectrometer (Agilent, Santa Clara, CA, USA) in a liquid chromatograph–electrospray ionization–quadrupole time-of-flight (LC-ESI-Q-TOF) setup (Figures S11–S16). The Agilent Technologies Cary 8454 UV-Vis spectrometer was employed to obtain UV/Vis absorption spectra in dichloromethane (CH$_2$Cl$_2$) solvent at 293 K. At 293 K, photoluminescence (PL) spectra were obtained using an Agilent Technologies Cary Eclipse Fluorescence spectrometer with dichloromethane (CH$_2$Cl$_2$) as the solvent. X-ray diffraction analysis (XRD) was conducted using Rigaku Ultima IV at room temperature.

### 2.3. Cyclic Voltammetry Measurements

Cyclic voltammetry (CV) experiments were conducted with a CHI 680D electrochemical analyzer using glassy carbon as the working electrode, Pt wire as the counter electrode and Ag/Ag$^+$ as the reference electrode using dichloromethane comprising 0.1 M tetrabutylammonium hexafluorophosphate as the solvent at a scan rate of 100 mV s$^{-1}$. Under these conditions, the reversible oxidation potential ($E_{1/2}$) of ferrocene was 0.40 V versus Ag/Ag$^+$ [51].

### 2.4. Computational Details

A GAUSSIAN 16 program was used to conduct all of the calculations [52]. Density functional theory computations using the PBE0 hybrid exchange-correlation functional completely optimized the geometries of the ground and excited states of Ir(III) complexes, with no symmetries being violated in the process [53]. The iridium atom was calculated using the effective core potentials of Hay and Wadt with a double f-valence basis set (LANL2DZ), whereas the rest of the atoms were calculated using the 6-311G(d,p) basis set [54]. Optimal geometries were used to determine vibrational frequencies, which were then used to confirm that the geometries represented potential energy minima. Furthermore, on the basis of the optimized ground-state geometries, the absorption characteristics in dichloromethane media were computed using TD-DFT at the CAM-B3LYP hybrid functional level and the PCM method [55]. It was determined that the stationary point was at minimum by completing frequency calculations at the same theoretical level (zero imaginary frequency). In every compound, the ligands were octahedrally coupled to the iridium metal.

### 2.5. Electrochemical Impedance Spectroscopy and Photocurrent Measurements

Electrochemical impedance spectroscopy (EIS) and photocurrent measurement were performed on CHI660E electrochemical workstation (Shanghai Chenhua, China), with Ag/AgCl as the reference electrode and Pt wire as the counter electrode, with a 0.5 M sodium sulphate (Na$_2$SO$_4$) solution as electrolyte. For the preparation of the working electrode, 10 mg of the iridium(III) dyes was dispersed in a 1 mL ethanol and 20 μL Nafion aqueous solution (5 wt%) and then ultrasonically scattered for 2 h. Subsequently, 0.1 mL of the slurry was added dropwise onto an FTO glass substrate (1 cm$^2$). The iridium(III) dye remained coated on the FTO glass surface after the ethanol had evaporated. The EIS was conducted at room temperature with a –700 mV bias potential and a 5 mV amplitude for the alternating current. The tested frequency range varied from 0.1 Hz to 100 kHz. Photocurrent measurements were produced throughout a frequency range of 0.1 Hz to 100 kHz. The illumination was provided by a 300 W Xe lamp. Everything was carried out at ambient temperature.

### 2.6. Preparation of Platinized TiO$_2$

Forty mL MeOH was added to a round bottom flask (r.b.f.) with 1.6 g of titanium oxide powder and 0.1 mL of aqueous hexachloroplatinic acid (H$_2$PtCl$_6$) solution (8 wt%). The reaction mixture was illuminated with a 300 W Hg lamp for 24 h. The mixture was then centrifuged at 4000 rpm for 5 min, and the solid formed was rinsed three times with methanol followed by drying under vacuum at room temperature in the dark overnight. Investigation of Pt-TiO$_2$ has been performed through a complete X-ray diffraction analysis. It is clear that the Pt-TiO$_2$ nanoparticles were properly manufactured, as shown by the XRD data (Figure S17) [56].

### 2.7. Adsorption of Iridium(III) Photosensitizer onto Platinized TiO$_2$

Twenty mg of platinized TiO$_2$ was placed into a centrifuge tube along with 2.5 mL of a 50 μM photosensitizer in CH$_2$Cl$_2$ solution, and the tube was then subjected to 30 min of sonication. The grey solid eventually transformed into dark pink as the solution became decolored and became transparent. Following centrifugation at 4000 rpm for 15 min, the liquid layer was removed and the resulting solid was dried overnight. Absorption peaks

in the low energy region of the spectra of the supernatant and the initial photosensitizer solution were compared to determine the dye-loading percentage.

### 2.8. Light-Driven Photocatalytic $H_2$ Production Studies

Five mL of 0.5 M AA (pH = 4) was added to a one-necked pear-shaped round bottom flask with Ir(III) dyes@Pt-TiO$_2$. Following 15 min of purging with an argon/methane (80:20 mol%) combination, the flask was sealed with a rubber septum. Methane in the gaseous mixture was used as an internal standard for the gas chromatograph (Agilent 6890 Series GC System with a 5 Å molecular-sieve column and thermal conductivity detector) analysis. A green (ca. 520 nm) or blue (ca. 470 nm) LED was placed in a properly sized container that blocked stray light from the surroundings, and the flask was churned constantly while emitting light from the bottom. The intensity of the illumination was evaluated with the aid of a thermal sensor and a power meter (Model: BIM-7203-0100F & BIM-7001; Hangzhou Brolight Technology Co., Ltd., Hangzhou, China), which was guesstimated as 50 mW. A gas chromatograph was used to quantify hydrogen generation from the reaction mixture's headspace at various time points, and a calibration plot showing the integrated values of hydrogen in comparison to the methane present was then used to determine the total amount of hydrogen generated (Figure S18). Based on the expected monochromaticity of the LED lights at their emission intensity maxima, approximations of the AQY% values were calculated.

$$AQY\% = \frac{rate\ of\ H_2\ production \times 2}{rate\ of\ incident\ photons} \times 100\%$$

### 2.9. Synthetic Procedure

**L1:** To a r.b.f. with 2-bromopyridine (0.8 mL, 8.549 mmol) in tetrahydrofuran (250 mL), (5-formylthiophen-2-yl)boronic acid (2.000 g, 12.824 mmol) was added. Tetrakis(triphenylphosphine)palladium(0) (0.988 g, 0.855 mmol) and 2M of K$_2$CO$_3$ (25.6 mL, 51.295 mmol) were added to the reaction mixture, which was then heated to 85 °C for 48 h. Upon cooling to room temperature, the mixture was subjected to extraction with ethyl acetate followed by a wash with brine. It was filtered and dried over sodium sulphate before being concentrated by reduced pressure. As a result of silica gel column chromatography using dichloromethane/$n$-hexane (1:1, $v/v$) as the eluent, the product was obtained as yellow solid (yield: 1.405 g, 88%). $^1$H NMR (400 MHz, CDCl$_3$) δ 9.95 (s, 1H), 8.66 (d, *J* = 4.8 Hz, 1H), 7.84−7.65 (m, 4H), 7.31 (d, *J* = 1.7 Hz, 1H). $^{13}$C NMR (101 MHz, CDCl$_3$) δ 183.2, 153.9, 151.1, 149.9, 144.1, 136.9, 125.1, 123.6, 119.7. Found: [M+H]$^+$ 190.0325; 'molecular formula C$_{10}$H$_7$NOS' requires [M+H]$^+$ 190.0321.

**L2:** To a flask comprising 1-chloroisoquinoline (1.400 g, 8.549 mmol) in tetrahydrofuran (250 mL), (5-formylthiophen-2-yl)boronic acid (2.000 g, 12.824 mmol) was added. Tetrakis(triphenylphosphine)palladium(0) (0.988 g, 0.855 mmol) and 2M of K$_2$CO$_3$ (25.6 mL, 51.295 mmol) were added to the reaction mixture, which was then heated to 115 °C for 48 h. The mixture was brought down to room temperature, then ethyl acetate and brine were used to extract the desired substances. An organic layer was concentrated after filtration and dried over sodium sulphate. The crude product was then concentrated by reduced pressure. A light-yellow solid was obtained when the crude product was refined through silica gel column chromatography with dichloromethane/$n$-hexane (1:1, $v/v$) as the eluent (yield: 1.737 g, 85%). $^1$H NMR (400 MHz, CDCl$_3$) δ 9.92 (d, *J* = 16.3 Hz, 1H), 8.32 (d, *J* = 8.3 Hz, 1H), 7.84 (d, *J* = 8.1 Hz, 1H), 7.77−7.63 (m, 5H), 7.59 (s, 1H). $^{13}$C NMR (101 MHz, CDCl$_3$) δ 183.0, 182.6, 141.4, 131.2, 128.5, 128.3, 126.9, 126.3, 120.8. Found: [M]$^+$ 239.0219; 'molecular formula C$_{14}$H$_9$NOS' requires [M]$^+$ 239.0399.

**Diethyl [2,2′-bipyridine]-4,4′-dicarboxylate**: [57] To a round bottom flask containing [2,2′-bipyridine]-4,4′-dicarboxylic acid (0.950 g, 3.893 mmol), concentrated H$_2$SO$_4$ (10.5 mL) and ethanol (22.5 mL) were added. The mixture was then heated to 80 °C overnight. Upon reaching ambient temperature, the mixture was poured on ice and neutralized to pH 8 with 2a 5% NaOH solution. Dichloromethane and brine were used to extract the mixture.

Sodium sulfate was used to dry the organic layer, which was then filtered and concentrated with reduced pressure. Toluene was utilized to recrystallize the crude product so as to make the final product a white solid (yield: 1.018 g, 87%). $^1$H NMR (400 MHz, CDCl$_3$) $\delta$ 8.96 (s, 2H), 8.89 (s, 2H), 7.92 (s, 2H), 4.47 (t, *J* = 10.6 Hz, 4H), 1.55−1.37 (m, 6H). $^{13}$C NMR (101 MHz, CDCl$_3$) $\delta$ 165.1, 156.5, 150.1, 138.9, 123.2, 120.5, 61.9, 14.2. Found: [M+Na]$^+$ 323.1045; 'molecular formula C$_{16}$H$_{16}$N$_2$O$_4$' requires [M+Na]$^+$ 323.1002.

**Ir1**: To a round bottom flask containing **L1** (0.500 g, 2.645 mmol), iridium(III) chloride hydrate (0.311 g, 0.881 mmol) were introduced with 2-ethoxyethanol: deionized water (3:1, *v/v*, total 8 mL). The reaction mixture was then heated up to 90 °C for 20 h. Through filtration, the mixture was refined to provide the yellow solid iridium dimer **Ir$_2$L1$_4$Cl**. This chemical was employed without additional purification in the following process.

To a round bottom flask containing iridium dimer **Ir$_2$L1$_4$Cl$_2$** (0.100 g, 0.083 mmol) in dichloromethane:methanol (1:1, *v/v*, total 6 mL), diethyl [2,2′-bipyridine]-4,4′-dicarboxylate (0.062 g, 0.208 mmol) was incorporated. The reaction mixture was then heated to 65 °C for 6 h. Upon cooling to room temperature, the pH was adjusted to 5 by introducing an appropriate amount of 1M HCl. The precipitate was filtered. Upon going through silica gel column chromatography with dichloromethane/methanol (1:1, *v/v*) as the eluent, the crude product was refined into a yellow solid (yield: 0.053 g, 37%). $^1$H NMR (400 MHz, CDCl$_3$) $\delta$ 9.20 (d, *J* = 5.4 Hz, 2H), 7.99−7.86 (m, 7H), 7.79 (dd, *J* = 11.4, 6.8 Hz, 6H), 7.60 (d, *J* = 7.9 Hz, 3H), 2.64 (s, 4H), 2.16 (s, 6H). Found: [M]$^+$ 869.1070; 'molecular formula C$_{36}$H$_{28}$IrN$_4$O$_6$S$_2$' requires [M]$^+$ 869.1073.

**Ir2**: To a round bottom flask containing iridium dimer **Ir$_2$L1$_4$Cl$_2$** (0.100 g, 0.083 mmol) in dichloromethane:methanol (1:1, *v/v*, total 6 mL), tetraethyl [2,2′-bipyridine]-4,4′-diylbis(phosphonate) (0.089 g, 0.208 mmol) was incorporated. A temperature of 65 °C was applied to the reaction mixture, and it was kept there for 6 h. Upon reaching ambient temperature, the pH was adjusted to 5 by introducing an appropriate amount of 1M HCl. The precipitate was filtered. A yellow solid was obtained when the crude product was refined using silica gel column chromatography with dichloromethane/methanol (1:1, *v/v*) as the eluent. (yield: 0.065 g, 39%). $^1$H NMR (400 MHz, CDCl$_3$) $\delta$ 9.94−9.47 (m, 3H), 9.20 (s, 2H), 7.84 (dd, *J* = 57.8, 50.0 Hz, 4H), 7.67−7.37 (m, 5H), 6.82−6.33 (m, 4H), 4.53−3.39 (m, 8H), 1.58−0.87 (m, 12H). Found: [M]$^+$ 997.1233; 'molecular formula C$_{38}$H$_{38}$IrN$_4$O$_8$P$_2$S$_2$' requires [M]$^+$ 997.1229.

**Ir3**: This product was synthesized using a similar approach as **Ir1**, in which **L1** was substituted with **L2** to obtain **Ir3** as a red solid. (yield: 0.066 g, 41%). $^1$H NMR (400 MHz, CDCl$_3$) $\delta$ 9.38 (d, *J* = 6.1 Hz, 2H), 7.71−7.50 (m, 10H), 7.04 (dd, *J* = 24.0, 5.8 Hz, 3H), 6.96−6.77 (m, 5H), 6.69 (dd, *J* = 19.7, 14.9 Hz, 2H), 2.94 (d, *J* = 29.5 Hz, 4H), 1.26 (d, *J* = 9.6 Hz, 6H). Found: [M]$^+$ 969.0983; 'molecular formula C$_{44}$H$_{32}$IrN$_4$O$_6$S$_2$' requires [M]$^+$ 969.1387.

**Ir4**: This product was prepared using a similar method as **Ir2**, except that **L1** was substituted with **L2**. **Ir4** was then obtained as a red solid. (yield: 0. 071 g, 39%). $^1$H NMR (400 MHz, CDCl$_3$) $\delta$ 9.17 (d, *J* = 6.2 Hz, 2H), 8.61 (s, 3H), 7.84 (dd, *J* = 12.0, 6.4 Hz, 6H), 7.61 (d, *J* = 6.6 Hz, 2H), 7.21 (d, *J* = 3.0 Hz, 6H), 7.02 (s, 3H), 3.76−3.58 (m, 8H), 2.37 (t, *J* = 7.5 Hz, 12H). Found: [M]$^+$ 1097.1545; 'molecular formula C$_{46}$H$_{42}$IrN$_4$O$_8$P$_2$S$_2$' requires [M]$^+$ 1097.1543.

## 3. Results and Discussion

### 3.1. Synthesis of the Materials

Schemes 1 and 2 demonstrate, respectively, the synthesis pathways for the C^N ligands and the iridium(III) photosensitizers. The iridium(III) photosensitizers are composed of an iridium(III) metal center with two C^N cyclometalating ligands (**L1** or **L2**) and either a [2,2′-bipyridine]-4,4′-dicarboxylate (**Ir1** and **Ir3**) or tetraethyl [2,2′-bipyridine]-4,4′-diylbis(phosphonate) (**Ir2** and **Ir4**) N^N auxiliary ligand. **L1** and **L2** were synthesized from a classic Suzuki reaction. Preparation of target iridium(III) dyes **Ir1**–**Ir4** involved two steps [58,59]. μ-chloride-bridged dimers were firstly synthesized by reacting the iridium(III)

chloride hydrate with the corresponding **L1** or **L2** CˆN cyclometalating ligand, which was then reacted with the NˆN auxiliary ligand to obtain the iridium(III) complexes (**Ir1**–**Ir4**). All of the iridium(III) complexes are stable as solids in air.

**Scheme 1.** Synthetic route for cyclometalating ligands **L1** and **L2** and diethyl [2,2′-bipyridine]-4,4′-dicarboxylate.

**Scheme 2.** Synthetic route for iridium(III) complexes **Ir1**–**Ir4**.

All of the organic precursors and ligands were measured by [1]H and [13]C NMR spectroscopy. A combination of LC-ESI-Q-TOF MS and [1]H NMR spectroscopy was used to analyze all of the iridium(III) complexes. A single peak at around ~9.95 ppm owes its existence to the proton of the aldehyde group [60]. The results provide a glimpse of the expected structures of the complexes. However, good quality acquisitions of their [13]C NMR spectra of were difficult due to their partial solubilities in common organic solvents.

### 3.2. Photophysical Properties

The photophysical properties of the iridium(III) complexes are explored in dichloromethane at room temperature. Their distinctive UV/Vis absorption spectra are recorded in Table 1, and a depiction of their spectra is provided in Figure 2. All of the dyes displayed strong absorption in the UV region at 290–320 nm, originating from intra-ligand charge transfer transition [61]. Spin-allowed $\pi$ to $\pi^*$ ligand-centered electronic transitions on both the C^N and N^N ligands provide the intense band at roughly 380 nm [31]. Metal-to-ligand charge transfer (MLCT) and ligand-to-ligand charge transfer (LLCT) were both thought to be responsible for the wide absorption bands above 400 nm [41]. These compounds showed higher absorption intensities in the visible region than the classical **[Ir(ppy)$_2$(dcbpy)]**$^+$ complex [27,28], which indicates that **Ir1–Ir4** had a better light-harvesting capability than that of **[Ir(ppy)$_2$(dcbpy)]**$^+$ [27,28,62]. In the range of 350–600 nm, the extended $\pi$-conjugation caused by the isoquinoline moiety in **Ir3** and **Ir4,** resulted in broader absorption peaks with higher $\varepsilon$ values than those of **Ir1** and **Ir2** [29,63]. Compared with **[Ir(ppy)$_2$(dcbpy)]**$^+$, the lowest energy MLCT band in all of the Ir(III) dyes was red-shifted [64], which can be explained by the extended $\pi$-conjugation in their C^N ligands [58,65]. Having a high $\varepsilon$ value in both the UV and visible light areas is widely considered to be a necessary characteristic of a photosensitizer for photocatalytic water-splitting hydrogen generation [58], and incorporating isoquinoline moiety in our dyes should enhance the photocatalytic hydrogen generation performance, as compared to classical iridium(III) dyes [66,67]. It is worth noting that the nature of the anchoring group only marginally affects the absorption properties of the iridium(III) dyes [44,68–73].

**Table 1.** UV/Vis absorption data of **Ir1−Ir4** in CH$_2$Cl$_2$ at 293 K.

| Photosensitizers | $\lambda_{max}$/nm ($\varepsilon$/10$^5$ M$^{-1}$ cm$^{-1}$) | $\lambda_{onset}$/nm |
|---|---|---|
| **Ir1** | 325(2.38), 418(0.69) | 550 |
| **Ir2** | 321(2.36), 414(0.67) | 539 |
| **Ir3** | 312(2.06), 360(1.78), 452(1.00), 492(0.71), 564(0.32) | 585 |
| **Ir4** | 308(2.03), 355(1.77), 447(0.98), 483(0.69), 553(0.31) | 576 |
| **[Ir(ppy)$_2$(dcbpy)]**$^+$ | 256(0.72), 308(0.32), 379(0.10) | 466 |

Photoluminescence spectra of **Ir1−Ir4** in a dichloromethane solution were measured at room temperature (Figure 3). Under photoexcitation at 450 nm, all of the Ir(III) dyes exhibited structured spectral emission ranging from 623 to 734 nm, resulting from the mixture of [3]MLCT and LC [3]$\pi$ to $\pi^*$ [74–76]. Significantly red-shifted emissions were observed in complexes **Ir3** and **Ir4,** as compared to **Ir1** and **Ir2**. This may be attributed to the extended $\pi$-conjugated in the cyclometalating ligands in **Ir3** and **Ir4** [31]. As shown by the PL data, changing the ligands efficiently tunes the photophysical characteristics.

### 3.3. Electrochemical Properties

In addition to the light-harvesting ability, it is crucial to have appropriate energy levels between the iridium(III) dyes, the semiconductor, and the SED for efficient hydrogen production. Efficient electron injection and charge separation occur when the LUMO of the iridium(III) dye has a higher energy level than the conduction band of the semicon-

ductor, and its HOMO has a lower energy level than the SED [77]. Table 2 illustrates the electrochemical characteristics of the dyes.

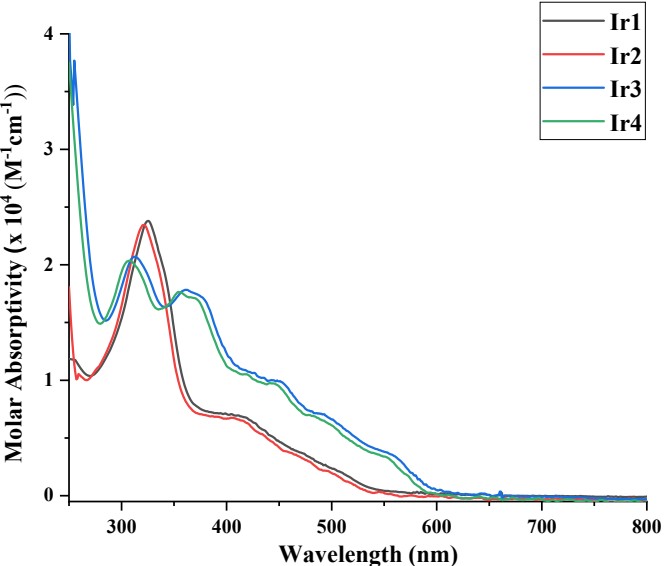

**Figure 2.** UV/Vis absorption spectra of **Ir1**−**Ir4** in $CH_2Cl_2$ solution at 293 K.

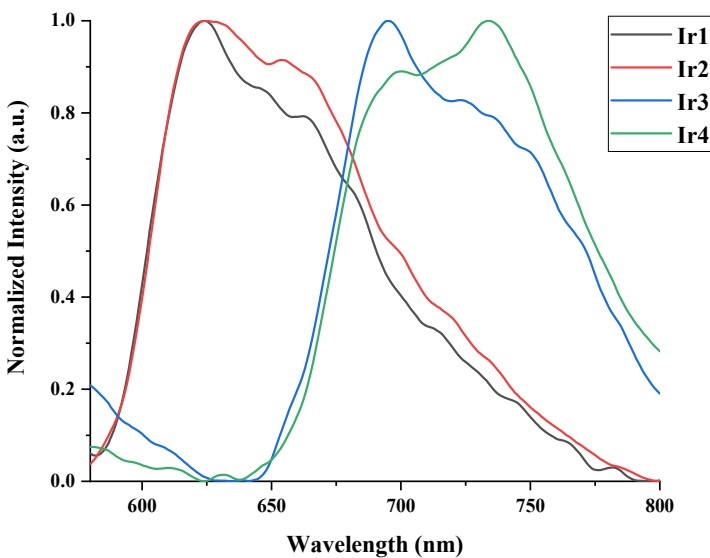

**Figure 3.** PL spectra of **Ir1**−**Ir4** in dichloromethane solution at room temperature.

**Table 2.** Electrochemical data and energy levels of **Ir1**−**Ir4**.

| Dye | $E_{Ox}^{Max}$/V | $E_{HOMO}$ [a]/eV | $E_g$ [b]/eV | $E_{ox*}$ [c]/V | $E_{LUMO}$ [d]/eV |
|-----|-----|-----|-----|-----|-----|
| **Ir1** | 0.76 | −5.56 | 2.25 | −1.49 | −3.31 |
| **Ir2** | 0.77 | −5.57 | 2.30 | −1.53 | −3.27 |
| **Ir3** | 0.73 | −5.53 | 2.12 | −1.38 | −3.41 |
| **Ir4** | 0.74 | −5.54 | 2.15 | −1.41 | −3.39 |

[a] Calculated from $-(E_{Ox}^{Max} + 4.8)$. [b] Energy bandgap ($E_g$) was determined from the onset of the absorption spectrum. [c] $E_{ox*} = E_{Ox}^{Max} - E_g$. [d] $E_{LUMO} = E_{HOMO} + E_g$.

The CV findings showed that semiconductor $TiO_2$'s conduction band (−4.4 eV) [78,79] was more negative than the $E_{LUMO}$ values of all four iridium(III) dyes (varying from −3.41

to −3.27 eV), allowing for effective electron injection in the photocatalytic water-splitting hydrogen production [80]. Meanwhile, the $E_{HOMO}$ values of **Ir1–Ir4** (–5.53 to –5.57 eV) are more negative than the redox potential level of the SED used in the water-splitting experiments (ascorbic acid; –4.65 eV at pH ~4), which ensured effective dye regeneration by the SED [80]. In **Ir2** and **Ir4**, the phosphonate groups provide strong chemical anchoring to the TiO$_2$ surface [42,44,69,81], which is attributed to the tuning of the LUMO energy level. These observations were corroborated via a density functional theory analysis.

### 3.4. Density Functional Theory Calculation

The density functional theory and time-dependent density functional theory (TD-DFT) calculations were carried out to elucidate the charge-transfer character and electronic properties of our complexes. The data are summarized in Table 3 and Figure 4. The TD-DFT calculations show that the HOMO→LUMO excitation is responsible for the majority of the electron's passage from the ground state (S$_0$) to the first singlet excited state (S$_1$). It is noteworthy that complexes sharing the same anchoring group have analogous LUMO energy levels, lending credence to the idea that LUMO energy levels are delocalized over the anchoring groups. As a consequence of photoexcitation-induced shifts in electron distribution, effective charge separation may be achieved. All of their electrochemical evidence agrees with these computer conclusions.

**Table 3.** Computational results of HOMO, LUMO, and energy gap (ΔE) of the Ir(III) dyes. The results are expressed in eV.

| Compounds | HOMO | LUMO |
|:---:|:---:|:---:|
| **Ir1** | −5.90 | −3.42 |
| **Ir2** | −5.90 | −3.33 |
| **Ir3** | −5.97 | −3.48 |
| **Ir4** | −5.98 | −3.38 |

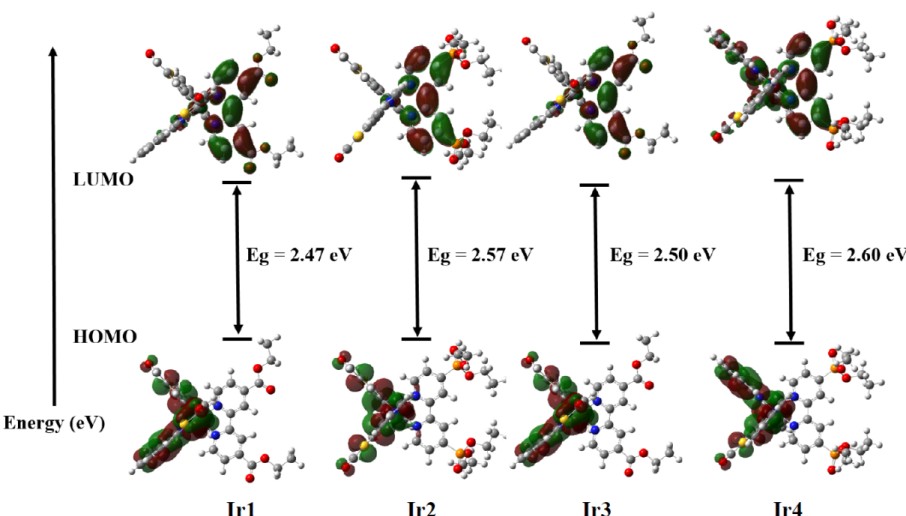

**Figure 4.** The calculated contour plots of the HOMO and LUMO energy levels of Ir(III) complexes.

### 3.5. Electrochemical Impedance Spectroscopy Measurements for Iridium(III) Dyes

Further exploration of the charge recombination characteristics of the iridium(III) complexes was performed using electrochemical impedance spectroscopy [82,83]. The evaluations were carried out using an electrochemical workstation [82,83]. Figure 5 displays the EIS Nyquist plot of our dyes. As is known to all, the smaller the arc radius in the EIS Nyquist plot, the lower the electric charge transfer resistance and the better the H$_2$ production performance [82–85]. Overall, the arc radii of **Ir4** was the smallest of all, which

suggests the best charge carrier transfer ability [86]. The results are in good agreement with the $H_2$ production performances of the dyes via water-splitting.

### 3.6. Photocurrent Measurements for Iridium(III) Dyes

Photocurrent measurement was used to test the stability and the relative effectiveness of charge separation of our metal complexes [87]. A uniform and fast photocurrent response during light-on/light-off testing indicates stable photocatalytic activity [88], while high photocurrent density indicates good charge separation efficiency [89]. Photocurrent measurements were conducted according to the previously reported procedure [90]. Figure 6 illustrates the photocurrent responses of **Ir1**–**Ir4** under visible light irradiation for six on-off cycles, which provides clear evidence of the electron transfer efficiency [90], thereby indicating their stable photocatalytic activities [88]. **Ir4** exhibited a significantly higher photocurrent intensity during light-on and decreased with a noticeable delay during light-off, thereby indicating its superior charge separation efficiency and better $H_2$ generation performance than the other three dyes [88,90–92].

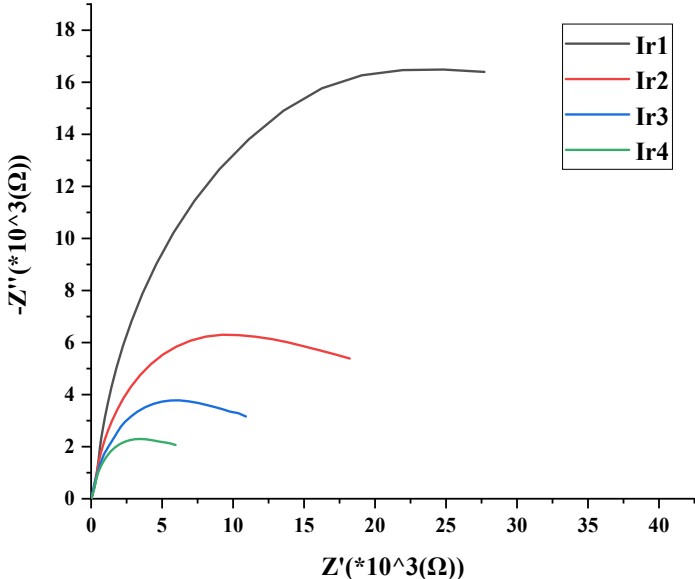

**Figure 5.** EIS Nyquist plot for **Ir1**–**Ir4**.

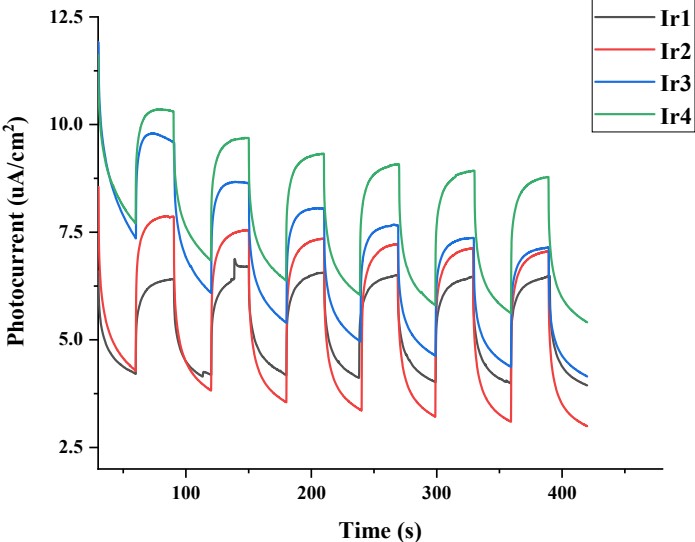

**Figure 6.** The photocurrent responses of **Ir1**–**Ir4**.

### 3.7. Light-Driven Hydrogen Generation Studies

In this photocatalytic experiment, iridium(III) dyes were used as photosensitizers in the photocatalytic hydrogen generation process through water-splitting. The experimental part includes instructions for making platinized $TiO_2$, attaching iridium(III) dyes to $Pt\text{-}TiO_2$, and its light-driven hydrogen reactions. Sonication was used to load each iridium(III) dye onto the $Pt\text{-}TiO_2$. Prior to performing the hydrogen generation experiment, the reaction mixture was centrifuged and allowed to dry. With reference to the different absorbance values prior to and following the dye-loading process, the dye-loading percentage (~100%) was ascertained for each sample (Figure S20).

The light-driven hydrogen production was carried out in 5 mL of an AA (0.5 M) solution at pH 4.0, with SED functioning as the reducing agent. Hydrogen was measured using GC with methane as an internal reference while the whole apparatus was constantly irradiated with green or blue LED. The hydrogen generation data for each water splitting system are presented in Figures 7 and 8, and the corresponding data are tabulated in Tables 4 and 5. In this configuration, light irradiation causes photoexcitation of the photosensitizer, subsequently leading to the infusion of electrons into the conduction band of $TiO_2$. To do this, they transfer their electrons to Pt nanoparticles doped into the $TiO_2$ surface, which in turn reduces protons to liberate hydrogen [31,58].

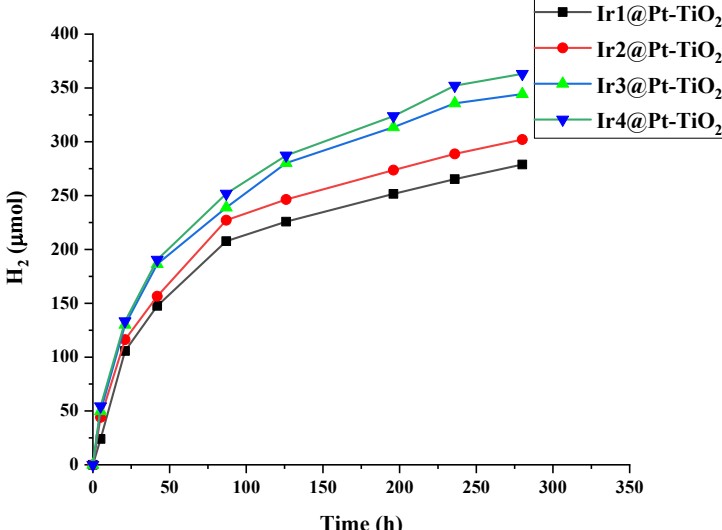

**Figure 7. Ir1–Ir4@Pt-TiO$_2$** photocatalytic $H_2$ production curves under illumination with blue LEDs (50 mW).

**Table 4.** Photocatalytic $H_2$ generation data with different iridium(III) dyes under blue light irradiation.

| Dye@Pt-TiO$_2$ | Time /h | H$_2$/mL | TON [a] | TOF [b]/h$^{-1}$ | TOF$_i$ [c]/h$^{-1}$ | Activity$_i$ [d] /μmol g$^{-1}$ h$^{-1}$ | AQY% | Dye Loading % |
|---|---|---|---|---|---|---|---|---|
| **Ir1** | 280 | 6.8 | 4462 | 15.9 | 76.9 | 48,052 | 0.91 | 100% |
| **Ir2** | 280 | 7.4 | 4834 | 17.3 | 141.1 | 88,163 | 0.98 | 100% |
| **Ir3** | 280 | 8.4 | 5509 | 19.7 | 159.5 | 99,673 | 1.13 | 100% |
| **Ir4** | 280 | 8.9 | 5809 | 20.7 | 174.0 | 108,735 | 1.19 | 100% |

[a] The TON value of $H_2$ was determined by multiplying the number of moles of $H_2$ produced by two and dividing it over the number of moles of Ir(III) dyes attached to $Pt\text{-}TiO_2$. [b] The TOF value was calculated per hour. [c] TOF$_i$ presents the initial TOF, which is calculated for the first 5 h. [d] Activity$_i$ is defined as the number of micromoles of $H_2$ evolved per gram of platinum loaded per hour.

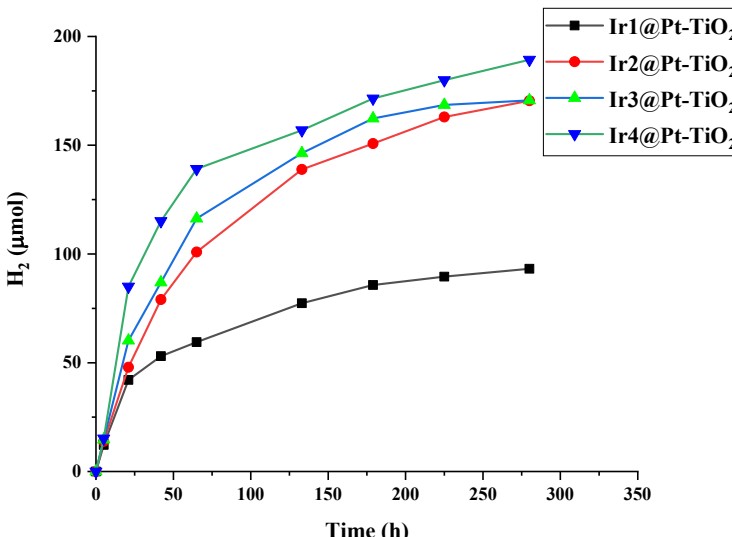

**Figure 8.** Ir1–Ir4@Pt-TiO₂ photocatalytic H₂ production curves under illumination with green LEDs (50 mW).

**Table 5.** Photocatalytic $H_2$ generation data with different iridium(III) dyes under green light irradiation.

| Dye@Pt-TiO₂ | Time /h | H₂/mL | TON [a] | TOF [b] /h⁻¹ | TOFᵢ [c] /h⁻¹ | Activityᵢ [d] /μmol g⁻¹ h⁻¹ | AQY% | Dye Loading % |
|---|---|---|---|---|---|---|---|---|
| **Ir1** | 280 | 2.3 | 1492 | 5.3 | 39.2 | 24,490 | 0.30 | 100% |
| **Ir2** | 280 | 4.2 | 2727 | 9.7 | 44.4 | 27,755 | 0.55 | 100% |
| **Ir3** | 280 | 4.2 | 2730 | 9.7 | 47.0 | 29,388 | 0.56 | 100% |
| **Ir4** | 280 | 4.6 | 3029 | 10.8 | 48.5 | 30,327 | 0.61 | 100% |

[a] The TON value of $H_2$ was determined by multiplying the number of moles of $H_2$ produced by two and dividing it over the number of moles of Ir(III) dyes attached to Pt-TiO₂. [b] The TOF value was calculated per hour. [c] TOFᵢ presents the initial TOF, which is calculated for the first 5 h. [d] Activityᵢ is defined as the number of micromoles of $H_2$ evolved per gram of platinum loaded per hour.

All four iridium(III) dye-based photocatalytic systems promoted hydrogen production via photocatalytic water splitting under irradiation with green or blue LEDs. In general, due to their superior light-harvesting capabilities in the blue light region, as compared to the green, the quantity of hydrogen generated under much higher levels of blue light radiation were emitted compared to green (495–570 nm) (Figure 2). Furthermore, all four dye-based systems generated hydrogen at a higher rate than the classical iridium-based dye **[Ir(ppy)₂(dcbpy)]⁺** under the same conditions (Figure S20 and Table S1). Their superior performance is possibly because of the improved light-harvesting ability over a wider range of wavelengths with an increased level of molar absorptivity ($\varepsilon$). The **Ir4** system exhibits the highest rates of hydrogen production in the presence of blue and green light, with 5809 and 3029 TON values, respectively. The system based on **Ir3** also exhibits satisfactory results, with TON values of 5509 and 2730 for blue and green light irradiation, respectively. The TON values of Ir(III) dyes show a descending order of **Ir4 > Ir3 > Ir2 > Ir1** when exposed to green and blue light. The results indicate that the polarizable π-system isoquinoline functional group effectively aids hydrogen production due to a better charge transfer ability [93,94]. In addition, iridium(III) dyes with phosphonate anchoring groups outperform those with carboxylate anchors (i.e., **Ir1** vs. **Ir2** and **Ir3** vs. **Ir4**), this probably due to the phosphonate anchoring groups that bind more strongly to the TiO₂ semiconductor [44,69]. The findings were consistent with those reported in the previous literature [42,81].

The **Ir4**@Pt-TiO₂ composite material continued to have its original color after radiation for 280 h (Figure S22), however, the AA aqueous solution's color transitioned from

colorless to a very light yellow owing to the existence of dehydroascorbic acid from the dye-regeneration process [58,95]. This demonstrates that the photocatalyst is not significantly affected by either dye desorption or photobleaching [96]. Taking into account the hydrogen production curve, it is fair to assume that the **Ir4**-based photocatalytic system would continue to function under much prolonged illumination.

## 4. Conclusions

Four new aldehyde-based iridium(III) complexes with either phosphonate or carboxylate anchoring groups are reported. All four iridium(III) photosensitizers are adequately characterized and their hydrogen productions via water-splitting were assessed. The initiation of the isoquinoline functional group in **Ir3** and **Ir4** effectively intensified and extended light absorption in the visible region, which are capable of exceptional light gathering and $H_2$ generation. A TON value of 5809 was achieved in the **Ir4**-based system under blue LED irradiation. In particular, iridium(III) dyes containing a phosphonate anchoring group were able to produce greater TON values than carboxylate-anchored dyes. Consequently, using phosphonate groups instead of carboxylate groups to bind iridium(III) dyes to a $TiO_2$ semiconductor provides a higher stability with more effective photocatalytic activity for photocatalytic $H_2$ generation from water. Future work related to those iridium(III) complexes may focus on employing these materials in biological applications, such as luminescent probes because of their attractive photophysical properties.

**Supplementary Materials:** The following supporting information can be downloaded at: https://www.mdpi.com/article/10.3390/inorganics11030110/s1, Figure S1: $^1$H NMR spectrum of L1 in CDCl$_3$; Figure S2: $^{13}$C NMR spectrum of L1 in CDCl$_3$; Figure S3: $^1$H NMR spectrum of L2 in CDCl$_3$; Figure S4: $^{13}$C NMR spectrum of L2 in CDCl$_3$; Figure S5: $^1$H NMR spectrum of diethyl [2,2′-bipyridine]-4,4′-dicarboxylate in CDCl$_3$; Figure S6: $^{13}$C NMR spectrum of diethyl [2,2′-bipyridine]-4,4′-dicarboxylate in CDCl$_3$; Figure S7: $^1$H NMR spectrum of **Ir1** in CDCl$_3$; Figure S8: $^1$H NMR spectrum of **Ir2** in CDCl$_3$; Figure S9: $^1$H NMR spectrum of **Ir3** in CDCl$_3$; Figure S10: $^1$H NMR spectrum of **Ir4** in CDCl$_3$; Figure S11: MS result of L1; Figure S12: MS result of L2; Figure S13: MS result of diethyl [2,2′-bipyridine]-4,4′-dicarboxylate; Figure S14: MS result of **Ir1**; Figure S15: MS result of **Ir2**; Figure S16: MS result of **Ir3**; Figure S17: MS result of **Ir4**; Figure S18: XRD patterns of Pt-TiO$_2$; Figure S19: Calibration plot of the signal ratio (H$_2$/CH$_4$) vs. amount of H$_2$ obtained from GC analysis; Figure S20: Cyclic voltammograms of **Ir1–Ir4**.; Figure S21: UV/Vis absorption spectra of **Ir1** to **Ir4** before and after dye loading in CH$_2$Cl$_2$ solution at 293 K; Figure S22: Photocatalytic H$_2$ generation curves of [Ir(ppy)$_2$(dcbpy)]$^+$ under blue LED irradiation (50 mW).; Table S1: Photocatalytic H$_2$ generation data with [Ir(ppy)$_2$(dcbpy)]$^+$ under blue light irradiation.

**Author Contributions:** X.Y.: contributed to the synthesis and characterization of iridium complexes, fabrication of light-driven hydrogen generation systems, wrote the article. Q.Z.: chemical formal analysis, discussed and revised the manuscript. P.-Y.H.: discussion. S.-C.Y.: discussion. S.S.: contributed to the DFT calculations. S.S.: contributed to the DFT calculations. S.H.: contributed to the DFT calculations. C.-L.H.: provided the methodology, wrote and revised the article. All authors have read and agreed to the published version of the manuscript.

**Funding:** C.-L. Ho thanks the Hong Kong Research Grants Council (PolyU 123021/17P), Environment and Conservation Fund (79/2020, P0034109) from the Government of HKSAR and the Hong Kong Polytechnic University (ZVVU and ZVXU) for their financial support. This research was also supported in part by the National Nanotechnology Center (NANOTEC), NSTDA, The Ministry of Higher Education, Science, Research and Innovation of Thailand, through its program of Research Network of NANOTEC (RNN).

**Data Availability Statement:** The data presented in this study are available on request from the corresponding author.

**Conflicts of Interest:** The authors declare no conflict of interest. The funders had no role in the design of the study; in the collection, analyses, or interpretation of data; in the writing of the manuscript; or in the decision to publish the results.

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
