# Peer review of "Development of Aldehyde Functionalized Iridium(III) Complexes Photosensitizers with Strong Visible-Light Absorption for Photocatalytic Hydrogen Generation from Water"

_inorganics, doi:10.3390/inorganics11030110_

Round 1

Reviewer 1 Report

This paper presents the design and synthesis of four iridium (III) complexes with anchoring units, describes the synthetic routes, and characterizes the electrochemical and photophysical properties of all the complexes. Density flooding theory (DFT) calculations were used to gain insight into the factors affecting the performance of Ir1-Ir4 as an effective photosensitizer for photocatalytic hydrogen production. It is an interesting well documented study but requires minor revisions by addressing the following points before publication.

1.     Section 2.4,2.5 It is recommended to write 2.4,2.5 together, so that the working electrode preparation does not need to be written twice.

2.     If the preparation of the material is involved, then the material needs to be characterized, i.e. the platinized titanium dioxide in the text needs XRD to support.

3.     Section 3.3 The CB value of titanium dioxide is mentioned, but the article does not mention the electrochemical test of titanium dioxide

4.     Section 3.3 Cyclic voltammetry tests need to be written out in the previous section

5.     The applied references are too old, it is recommended to read more articles from recent years

6.     The vertical coordinate units in Figure 6 and Figure 7 are not common and should be referred to other literature

Reviewer 2 Report

In current work (Manuscript ID: inorganics-2233640), authors report Four iridium(III) dyes functionalized with aldehyde moiety in their cyclometalating (C^N) ligands; bearing either diethyl [2,2'-bipyridine]-4,4'-dicarboxylate or tetraethyl[2,2'-bipyri-dine]-4,4'-diylbis(phosphonate) anchoring groups; coded as Ir1–Ir4; and explored as photosensitizers. The complexes are characterized with respect to their electrochemical and photophysical properties. Density functional theory (DFT) cal-culation was used to gain insight into the factors responsible for photocatalytic properties of Ir1–Ir4 as effective photosensitizers for photocatalytic hydrogen generation. Relative to common irid-ium(III) dyes; such as [Ir(ppy)2(dcbpy)]+ (ppy = 2-phenylpyridine); the absorption spectra of dyes are broader; which is attributed to the extended π-conjugation in their C^N ligands. All the new iridium(III) dyes were used as photosensitizers for visible-light driven hydrogen production by attaching to platinized TiO2 nanoparticles (Pt–TiO2) in the presence of sacrificial electron donor (SED) of ascorbic acid (AA) in purely aqueous solution. A H2 turnover number (TON) up to 5809 was demonstrated for 280h irradiation. Complexes with tetraethyl[2,2′-bipyridine]-4,4′-diylbis(phosphonate) anchoring groups were found to be outperform to those with classical diethyl [2,2'-bipyridine]-4,4'-dicarboxylate; which may be one of the crucial steps in designing superior iridium (III) photosensitizers for hydrogen generation from pure water. This work has serious issues. Therefore, I would not consider its publication in inorganics.

1. Some typos errors and grammatical mistakes exist throughout. Such as, at the end of the last sentence of the abstract part punctuation mark is needed.

2. The introduction part needs further improvement.

3. The authors mentioned that to estimate AQY%, LED lights with single wavelengths emission intensity (i.e., 470 and 520 nm) were used. The authors didn’t mention the duration of the photocatalytic reaction for estimating the AQY.

4. In Figure 2, the Black and Red, and Blue and Dark Cyan spectra seem to be exact copies of each other. Authors should remeasure UV/Vis absorption spectra.

5. In Figure 5, why the baseline of the photocurrent responses is not at the same position? It seems that the spectra have been shifted upside.

6. There are no specific points of time intervals in the H2 evolution graphs.

Reviewer 3 Report

Yao and co-workers have developed new Iridium (III) complex based photosensitizers and studied their optical and electrochemical properties.  Further, they have applied these complexes for photocatalytic hydrogen generation from water. The results presented are new and can be published. However, the authors should first address the concerns, as mentioned below:

1. Photoluminiscence quenching of the iridum complexes films adsorbed on TiO2 is an effective way to show how efficient is the electron injection into TiO2. The authors should measure the PL quenching for the four complexes synthesized and include in the manuscript.

2. The authors should include the stability data of the iridium complexes after hours of irradiation, for example, for 280 h irradiation.

3. Did the authors observe any desorption of the iridium complexes from the TiO2 surface after the experiments?

Round 2

Reviewer 2 Report

Authors have well addressed the reviewers comments. The manuscript can be accepted in it's current form.